# Enhancing Linear Attention with Residual Learning

## Abstract

Linear attention offers a linear-time alternative to self-attention but often struggles to capture long-range patterns. We revisit linear attention through a prediction-correction lens and show that prevalent variants can be written as a combination of a historical prediction and a single-token correction, which creates an expressivity bottleneck. To address this bottleneck, we introduce **Residual Linear Attention (RLA)**, a framework that equips linear attention with an explicit *residual-fitting* mechanism. RLA maintains an auxiliary recurrent state that learns to accumulate residual errors over time and correct the base prediction. We further instantiate a delta-rule version, **Residual Delta Net (RDN)**, incorporating adaptive gating and residual clipping for enhanced correction control and stability. Our implementation leverages highly optimized linear attention kernels and preserves linear time and memory. Across language modeling and recall-intensive evaluations, RLA and RDN consistently outperform their respective baselines and other modern linear-attention methods, narrowing the gap to standard Transformers while retaining linear scaling.

## 1 Introduction

The Transformer (Vaswani et al., 2017) architecture has become the standard for large language models. However, the quadratic time complexity of its self-attention mechanism remains a critical bottleneck, limiting its application to long sequences (Li et al., 2024). Linear attention has recently emerged as an efficient alternative to standard self-attention, directly addressing its prohibitive quadratic complexity. By reformulating the attention computation into a recurrent process, these models achieve linear-time training and inference, making them well-suited for processing long sequences. Architectures such as RetNet (Sun et al., 2023) and Mamba (Gu & Dao, 2023; Dao & Gu, 2024) have demonstrated competitive performances. Methods like GLA (Yang et al., 2023) and DeltaNet (Yang et al., 2024b) offer further improvements by incorporating data-dependent gating and state update rules to manage the flow of information within a single state matrix.

Modern linear attention methods can be unified as learning a direct mapping from keys to values (Sun et al., 2024), a process analogous to test-time training. For example, the delta update rule (Schlag et al., 2021) can be derived from a single step of online gradient descent on a quadratic loss objective. This perspective opens several avenues for improvement. These include exploring different online learning loss functions to derive new update rules (Schlag et al., 2021; Yang et al., 2024b), employing more sophisticated mapping functions, or modifying the online gradient update mechanism (von Oswald et al., 2025; Siems et al., 2025). For instance, recent works like TTT-MLP (Sun et al., 2024) and Titans (Behrouz et al., 2024) utilize a Multi-Layer Perceptron (MLP) as a deep memory module to achieve a more powerful mapping. However, this approach sacrifices the model's linear recurrence, thereby complicating parallel training.

Building on this perspective, we offer a new interpretation of the attention output. We show that the output of prevalent linear attention models can be decomposed into a base component generated from historical states and a correction term derived solely from the current token (see Section 2.3). Relying on a single token to perform this systematic correction imposes a bottleneck and is detrimental to the model's expressive power. To address these issues, we introduce Residual Linear Attention, a framework that enhances linear attention models with an explicit residual fitting mechanism. Rather than depending on a single token for correction, our method employs an auxiliary state

matrix to explicitly model and correct systematic prediction errors of the base linear attention. The final output is a combination of the base prediction and this learned error correction. Our approach can be generalized to a unified framework applicable to various linear attention methods, offering a powerful and efficient strategy for building more capable sequence models.

Building upon existing linear attention methods, we propose two variants enhanced with residual fitting: Residual Linear Attention (RLA) and Residual Delta Net (RDN). We evaluate them on a range of benchmarks, including language modeling and recall-intensive tasks. Our results demonstrate that these models outperform their respective baselines and other modern linear attention methods, while our ablation analysis confirms the importance of each key design choice within our framework.

## 2 PRELIMINARIES

### 2.1 LINEAR ATTENTION AS A RECURRENT MODEL

Softmax attention mechanisms exhibit quadratic computational complexity with respect to sequence length, constituting a significant bottleneck when processing long sequences. Linear attention (Katharopoulos et al., 2020) architectures address this by removing the softmax function, which allows for a reordering of the computation.

For the $t$-th token in a sequence, let the query, key, and value vectors be $\boldsymbol{q}_t \in \mathbb{R}^{d_q \times 1}$, $\boldsymbol{k}_t \in \mathbb{R}^{d_k \times 1}$, and $\boldsymbol{v}_t \in \mathbb{R}^{d_v \times 1}$, where $d_q$, $d_k$, and $d_v$ are their respective feature dimensions, with $d_q = d_k$. After applying a kernel function $\phi(\cdot)$ to the queries and keys (omitted for simplicity in the notation), the causal linear attention output $\boldsymbol{o}_t$ can be expressed as:

$$\boldsymbol{o}_t = \sum_{i=1}^{t} \boldsymbol{v}_i \left( \boldsymbol{k}_i^\top \boldsymbol{q}_t \right) = \left( \sum_{i=1}^{t} \boldsymbol{v}_i \boldsymbol{k}_i^\top \right) \boldsymbol{q}_t .$$

By defining a state matrix $\boldsymbol{S}_t := \sum_{i=1}^{t} \boldsymbol{v}_i \boldsymbol{k}_i^\top \in \mathbb{R}^{d_v \times d_k}$, we arrive at the following recurrent formulation:

$$\boldsymbol{S}_t = \boldsymbol{S}_{t-1} + \boldsymbol{v}_t \boldsymbol{k}_t^\top, \quad \boldsymbol{o}_t = \boldsymbol{S}_t \boldsymbol{q}_t .$$

This recurrent form maintains constant time and memory complexity per step during inference and facilitates efficient training through chunk-wise parallel algorithms (Yang et al., 2023). Furthermore, the use of gating mechanisms has led to the development of more variants such as RetNet (Sun et al., 2023), Lightning Attention (Qin et al., 2024a), and Mamba-2 (Dao & Gu, 2024).

### 2.2 AN ONLINE LEARNING PERSPECTIVE

The design of the recurrent update rule can be motivated from an online learning perspective (Sun et al., 2024; Liu et al., 2024). In this view, the token sequence is a stream of data points $(\boldsymbol{k}_t, \boldsymbol{v}_t)$, and the state matrix $\boldsymbol{S}$ acts as model parameters. These parameters are updated online to learn the mapping $\boldsymbol{k} \mapsto \boldsymbol{v}$, with $\boldsymbol{S}$ functioning as a memory from which information is retrieved using the query $\boldsymbol{q}$ via $\boldsymbol{S}\boldsymbol{q}$.

The state update can be interpreted as one step of gradient descent on a loss function $\mathcal{L}(\boldsymbol{k}, \boldsymbol{v}; \boldsymbol{S})$. For instance, applying a single descent step with the loss $\mathcal{L}(\boldsymbol{k}_t, \boldsymbol{v}_t; \boldsymbol{S}) := -\langle \boldsymbol{S}\boldsymbol{k}_t, \boldsymbol{v}_t \rangle$ recovers the standard linear attention update:

$$\boldsymbol{S}_t = \boldsymbol{S}_{t-1} - \nabla_{\boldsymbol{S}} \mathcal{L}(\boldsymbol{k}_t, \boldsymbol{v}_t; \boldsymbol{S}_{t-1}) = \boldsymbol{S}_{t-1} + \boldsymbol{v}_t \boldsymbol{k}_t^\top .$$

An alternative update rule can be derived by minimizing a squared error loss, $\mathcal{L}(\boldsymbol{k}_t, \boldsymbol{v}_t; \boldsymbol{S}) := \frac{1}{2} \|\boldsymbol{S}\boldsymbol{k}_t - \boldsymbol{v}_t\|^2$. Performing one step of gradient descent on $\boldsymbol{S}_{t-1}$ with a data-dependent learning rate $\beta_t$ yields the delta rule:

$$\boldsymbol{S}_t = \boldsymbol{S}_{t-1} - \beta_t \nabla_{\boldsymbol{S}} \mathcal{L}(\boldsymbol{k}_t, \boldsymbol{v}_t; \boldsymbol{S}_{t-1}) = \boldsymbol{S}_{t-1}(I - \beta_t \boldsymbol{k}_t \boldsymbol{k}_t^\top) + \beta_t \boldsymbol{v}_t \boldsymbol{k}_t^\top .$$

This formulation enables models like Delta Net (Yang et al., 2024b; Schlag et al., 2021) to achieve fine-grained memory control. Gated Delta Net (Yang et al., 2024a) further enhances this approach by incorporating weight decay into the learning process.

## 2.3 Decomposition into Prediction and Correction

We interpret linear attention through a prediction-correction lens. The standard linear attention output, $o_t = S_t q_t$, can be viewed as the sum of a base prediction from the past state and a correction based on the current token:

$$o_t = \underbrace{S_{t-1} q_t}_{\text{Base Prediction}} + \underbrace{\left(v_t k_t^\top\right) q_t}_{\text{Error Correction}}.$$

We can generalize this decomposition to the form $o_t = S_{t-1} q_t + R_t q_t$, where we introduce $R_t$ as a generalized correction state. This framework provides a unified view of several methods, as shown in Table 1, which differ not only in the design of their associated state update but also in this correction term.

Table 1: Comparison of different linear attention methods with base prediction and error correction.

| Method | Output Combination | State Update Rule | Correction Term |
|---|---|---|---|
| LinearAttn | $o_t = S_{t-1} q_t + R_t q_t$ | $S_t = S_{t-1} + v_t k_t^\top$ | $R_t = v_t k_t^\top$ |
| Mamba2 | $o_t = \alpha_t S_{t-1} q_t + R_t q_t$ | $S_t = \alpha_t S_{t-1} + v_t k_t^\top$ | $R_t = v_t k_t^\top$ |
| Gated LinearAttn | $o_t = \text{diag}(\alpha_t) S_{t-1} q_t + R_t q_t$ | $S_t = \text{diag}(\alpha_t) S_{t-1} + v_t k_t^\top$ | $R_t = v_t k_t^\top$ |
| DeltaNet | $o_t = S_{t-1} q_t + \beta_t R_t q_t$ | $S_t = S_{t-1}(I - \beta_t k_t k_t^\top) + \beta_t v_t k_t^\top$ | $R_t = (v_t - S_{t-1} k_t) k_t^\top$ |
| Gated DeltaNet | $o_t = \alpha_t S_{t-1} q_t + \beta_t R_t q_t$ | $S_t = \alpha_t S_{t-1}(I - \beta_t k_t k_t^\top) + \beta_t v_t k_t^\top$ | $R_t = (v_t - \alpha_t S_{t-1} k_t) k_t^\top$ |

Building on the prediction-correction viewpoint, we introduce a residual fitting framework to enhance linear attention. Our framework learns a more expressive correction term by explicitly fitting on contextual information beyond the current token.

## 3 Method

This section presents our proposed method, which enhances linear attention through a residual-fitting process. We begin by describing the foundational residual learning framework that underpins our method. Next, we introduce an adaptive correction factor to enhance modeling capabilities and clipping methods to stabilize the residual fitting process. Finally, we present two final variants of our approach.

### 3.1 Explicit Residual Fitting

As established in Section 2.3, the output of linear attention can be decomposed into a base prediction and a correction term. To learn a more expressive correction, we introduce an auxiliary state, $R_t$, which modifies the output formulation to $o_t = S_{t-1} q_t + R_t q_t$. Crucially, unlike the standard correction shown in Table 1, which is derived solely from the current token, our auxiliary state $R_t$ is updated recurrently, analogous to the primary state $S_t$.

The learning target for state $R_t$ is motivated by a second-order analysis of the loss function. Given a prediction $\hat{v}$, the Taylor expansion of a loss function $\mathcal{L}(\hat{v}, v)$ around $\hat{v}$ with a small perturbation $\delta$ is:

$$\mathcal{L}(\hat{v} + \delta, v) \approx \mathcal{L}(\hat{v}, v) + (\nabla_{\hat{v}} \mathcal{L})^\top \delta + \frac{1}{2} \delta^\top \left(\nabla_{\hat{v}}^2 \mathcal{L}\right) \delta.$$

Minimizing this approximation with respect to $\delta$ suggests an optimal update step, $\delta^* = -\left(\nabla_{\hat{v}}^2 \mathcal{L}\right)^{-1} (\nabla_{\hat{v}} \mathcal{L})$. For the commonly used L2 loss, $\mathcal{L} = \frac{1}{2}\|v - \hat{v}\|^2$, this optimal update simplifies directly to the residual error, $r := \delta^* = v - \hat{v}$. This motivates modeling the residual with our auxiliary state, which we define as $r_t := v_t - S_{t-1} k_t$.

Leveraging the online learning perspective of linear attention from Section 2, we apply an analogous update rule to the auxiliary state. This yields the following recurrent process:

$$\boldsymbol{r}_t = \boldsymbol{v}_t - \boldsymbol{S}_{t-1}\boldsymbol{k}_t \qquad \text{(Residual Error Computation)}$$

$$\boldsymbol{R}_t = \boldsymbol{R}_{t-1} + \boldsymbol{r}_t\boldsymbol{k}_t^\top \qquad \text{(Auxiliary State Update)}$$

$$\boldsymbol{o}_t = \boldsymbol{S}_{t-1}\boldsymbol{q}_t + \boldsymbol{R}_t\boldsymbol{q}_t \qquad \text{(Output Combination)}$$

$$\boldsymbol{S}_t = \boldsymbol{S}_{t-1} + \boldsymbol{v}_t\boldsymbol{k}_t^\top \qquad \text{(Base State Update)}$$

In this formulation, the auxiliary state $\boldsymbol{r}_t$ accumulates past residual errors and their corresponding keys. This allows it to model and correct for systematic prediction errors made by the base state $\boldsymbol{S}_{t-1}$, yielding a more expressive output. Furthermore, we generalize this residual fitting process to formulate a unified framework for boosting linear attention, with a detailed derivation provided in Appendix A.

It is worth noting two special cases of this formulation. If we set $\boldsymbol{R}_t = \boldsymbol{v}_t\boldsymbol{k}_t^\top$, using information only from the current token, our method reduces to standard linear attention (Katharopoulos et al., 2020). If we instead use $\boldsymbol{R}_t = (\boldsymbol{v}_t - \boldsymbol{S}_{t-1}\boldsymbol{k}_t)\boldsymbol{k}_t^\top$, the correction mechanism becomes equivalent to a one-step delta rule update (Schlag et al., 2021).

## 3.2 Adaptive Gating and Correction Factor

To enhance control over the state dynamics, we incorporate learnable gating scalars, a practice common in recent recurrent models (Yang et al., 2024a; Dao & Gu, 2024). We introduce a decay factor $\alpha_t \in [0, 1]$ to control the retention of past information, and an update rate $\beta_t \in [0, 1]$ to modulate the influence of the current token. These factors can be applied to both state updates and output combinations:

$$\boldsymbol{S}_t = \alpha_t\boldsymbol{S}_{t-1} + \beta_t\boldsymbol{v}_t\boldsymbol{k}_t^\top$$

$$\boldsymbol{R}_t = \alpha_t\boldsymbol{R}_{t-1} + \beta_t\boldsymbol{r}_t\boldsymbol{k}_t^\top$$

$$\boldsymbol{o}_t = \alpha_t\boldsymbol{S}_{t-1}\boldsymbol{q}_t + \beta_t\boldsymbol{R}_t\boldsymbol{q}_t$$

However, using the same update rate $\beta_t$ for both states couples the learning of the base representation and the error correction. To achieve more fine-grained control, we introduce a dedicated scalar correction factor, $\gamma_t \in [0, 1]$. This factor decouples the update processes and allows the model to dynamically scale the contribution of the residual correction term. The auxiliary state updates and output computation are given by:

$$\boldsymbol{S}_t = \alpha_t\boldsymbol{S}_{t-1} + \beta_t\boldsymbol{v}_t\boldsymbol{k}_t^\top$$

$$\boldsymbol{R}_t = \alpha_t\boldsymbol{R}_{t-1} + \gamma_t\boldsymbol{r}_t\boldsymbol{k}_t^\top$$

$$\boldsymbol{o}_t = \alpha_t\boldsymbol{S}_{t-1}\boldsymbol{q}_t + \gamma_t\boldsymbol{R}_t\boldsymbol{q}_t$$

This formulation uses the decay and correction factors to dynamically gate the retrieval from the base and auxiliary states, respectively.

## 3.3 Normalization and Residual Clipping

To ensure computational stability, we introduce two mechanisms. First, we apply L2 normalization to the query and key vectors to improve numerical stability. Second, we address potential instability in the auxiliary state $\boldsymbol{r}_t$ by clipping the residual:

$$\boldsymbol{r}_t = \text{Clip}_{[-c,c]}(\boldsymbol{v}_t - \boldsymbol{S}_{t-1}\boldsymbol{k}_t).$$

This ensures that the error-correction state $\boldsymbol{r}_t$ maintains a stable learning trajectory, even when the base model produces transient, large prediction errors. A detailed derivation for this clipping method is provided in Appendix B.

## 3.4 FINAL FORMULATIONS

The residual fitting principle is a general technique that can be integrated with various linear attention backbones. By applying our residual mechanism to both the standard additive update rule and the delta update rule, we derive two powerful variants. This leads to our final models:

$$
\begin{aligned}
\boldsymbol{r}_t &= \mathrm{Clip}_{[-c,c]}(\boldsymbol{v}_t - \boldsymbol{S}_{t-1}\boldsymbol{k}_t) \\
\boldsymbol{R}_t &= \alpha_t \boldsymbol{R}_{t-1} + \gamma_t \boldsymbol{r}_t \boldsymbol{k}_t^\top \\
\boldsymbol{S}_t &= \alpha_t \boldsymbol{S}_{t-1} + \beta_t \boldsymbol{v}_t \boldsymbol{k}_t^\top \\
\boldsymbol{o}_t &= \alpha_t \boldsymbol{S}_{t-1}\boldsymbol{q}_t + \gamma_t \boldsymbol{R}_t \boldsymbol{q}_t
\end{aligned}
\qquad
\begin{aligned}
\boldsymbol{r}_t &= \mathrm{Clip}_{[-c,c]}(\boldsymbol{v}_t - \boldsymbol{S}_{t-1}\boldsymbol{k}_t) \\
\boldsymbol{R}_t &= \alpha_t \boldsymbol{R}_{t-1}(I - \gamma_t \boldsymbol{k}_t \boldsymbol{k}_t^\top) + \gamma_t \boldsymbol{r}_t \boldsymbol{k}_t^\top \\
\boldsymbol{S}_t &= \alpha_t \boldsymbol{S}_{t-1}(I - \beta_t \boldsymbol{k}_t \boldsymbol{k}_t^\top) + \beta_t \boldsymbol{v}_t \boldsymbol{k}_t^\top \\
\boldsymbol{o}_t &= \alpha_t \boldsymbol{S}_{t-1}\boldsymbol{q}_t + \gamma_t \boldsymbol{R}_t \boldsymbol{q}_t
\end{aligned}
$$

**Residual Linear Attention (RLA)**            **Residual Delta Net (RDN)**

For brevity, the equations omit the L2 normalization and SiLU activation applied to query and key vectors. Regarding the adaptive gates, the decay factor $\alpha_t$ adopts the re-parameterization from Mamba-2 (Dao & Gu, 2024), while $\beta_t$ and $\gamma_t$ are computed via a linear projection followed by a sigmoid activation. The structure of our attention block is shown in Figure 1.

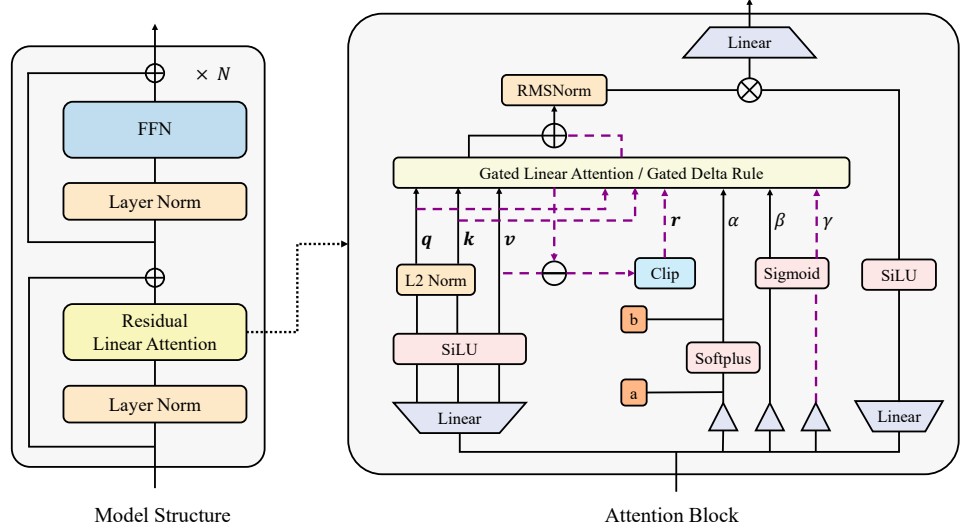

Model Structure            Attention Block

Figure 1: The architecture of our proposed model. The model structure (left) consists of $N$ stacked blocks. The detailed Attention Block (right) illustrates our core mechanism. Our primary contribution, the explicit residual fitting process, is highlighted in purple dash lines. This path computes the clipped residual $\boldsymbol{r}_t = \mathrm{Clip}(\boldsymbol{v}_t - \boldsymbol{S}_{t-1}\boldsymbol{k}_t)$, which is then modulated by a dedicated correction factor $\gamma_t = \sigma(\boldsymbol{W}_\gamma \boldsymbol{x})$ to dynamically correct the base prediction from the model's primary state. The model also utilizes gates $\alpha_t = \exp(-a\,\mathrm{softplus}(\boldsymbol{W}_\alpha \boldsymbol{x} + b))$ and $\beta_t = \sigma(\boldsymbol{W}_\beta \boldsymbol{x})$ to control the state dynamics, where $a$ and $b$ are learnable scalars.

## 4 EXPERIMENT

### 4.1 SETUP

**Implementation** To maximize efficiency, we implement our custom attention kernels in Triton (Tillet et al., 2019), building upon the `flash-linear-attention` library (Yang & Zhang, 2024). We exploit the fact that our state update rule is identical to linear attention's, requiring only a minor modification to their kernel: we augment it to return both the attention result and the intermediate residual. This design allows the same highly optimized kernel to be reused across all residual-fitting stages, ensuring high throughput.

**Model Settings** We evaluate our model against several recent linear attention architectures, including Retentive Network (RetNet) (Sun et al., 2023), Mamba2 (Dao & Gu, 2024), and Gated Delta Net (GDN) (Yang et al., 2024a). Additionally, we establish a baseline for RLA by evaluating scalar-gated linear attention (sGLA), a linear attention variant equipped with query-key normalization and scalar gates ($\alpha$ and $\beta$). In our main experiments, we set the clipping threshold to $c = 1$. All models contain approximately $1.5$ billion parameters and are trained on $100$ billion tokens under identical conditions to ensure a fair comparison. Further details on the training configuration can be found in Appendix C.

## 4.2 MAIN RESULTS

**Kernel Efficiency** We benchmark our kernel's runtime against linear attention baselines and FlashAttention (Dao et al., 2022; Dao, 2023), as shown in Figure 2. Although the residual fitting process adds computational overhead, our method's runtime scales linearly with sequence length. This makes it significantly faster than FlashAttention, which scales quadratically, on longer sequences. Regarding throughput, our method, like other linear attention mechanisms, maintains a nearly constant high throughput. Conversely, the throughput of the compute-bound FlashAttention degrades rapidly as sequence length increases.

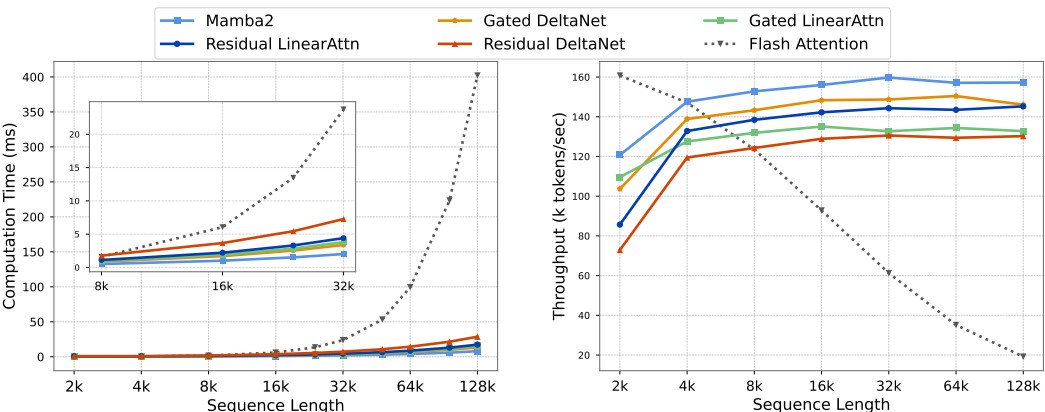

Figure 2: Comparison of attention kernel computation time (left) and model throughput (right) with respect to sequence length.

**Language Modeling & Commonsense Reasoning** We evaluate RLA and RDN on Wiki-Text (Merity et al., 2016) perplexity and a suite of benchmarks assessing reasoning and commonsense understanding. The reasoning tasks include ARC-Easy, ARC-Challenge (Clark et al., 2018), PIQA (Bisk et al., 2020), and MMLU (Hendrycks et al., 2020), while commonsense understanding is evaluated on HellaSwag (Zellers et al., 2019), Winogrande (Sakaguchi et al., 2021), SocialIQA (Sap et al., 2019), and LAMBADA (Paperno et al., 2016). Our main results, summarized in Table 2, show that our proposed residual learning variants, RLA and RDN, achieve a consistent improvement in perplexity over their respective baselines, sGLA and GDN. In addition, our models outperform other leading linear attention methods across multiple benchmarks and deliver performance competitive with a standard Transformer.

**Recall-intensive tasks** To evaluate memory capacity, we benchmark our model on the recall-intensive tasks from Arora et al. (2024). In addition, we also directly evaluate the model's retrieval ability using the "Needle-in-a-Haystack" task (NIAH) (gkamradt, 2023), which requires retrieving key-value pairs inserted at varying depths within a long document. These benchmarks are challenging for linear attention models because their finite state-space creates an information bottleneck, as shown in Table 3. Results demonstrate that our proposed RLA and RDN consistently outperform their corresponding baselines, with particularly strong gains on the DROP and FDA benchmarks. Furthermore, they substantially outperform other models on the NIAH task, highlighting an enhanced capacity for information recall.

Table 2: Language modeling, reasoning, and commonsense understanding results. We report perplexity (lower is better) and accuracy (higher is better). The **bold**/underlined numbers indicate the first/second best values of the linear attention models in each column.

| Model | Wiki ppl | LAMB ppl | ARC-C acc_n | ARC-E acc | HellaSwag acc_n | LAMB acc | MMLU acc_n | PIQA acc | SIQA acc | Wino acc | Avg Acc |
|---|---|---|---|---|---|---|---|---|---|---|---|
| Transformer | 17.33 | 19.53 | 30.2 | 55.7 | 49.0 | 44.3 | 31.1 | 70.4 | 37.9 | 53.5 | 46.51 |
| RetNet | 18.86 | 27.62 | 29.4 | 56.0 | 40.5 | 36.5 | 30.3 | 69.9 | 36.0 | 50.9 | 43.69 |
| Mamba2 | 18.42 | 20.80 | 29.1 | 57.4 | 46.2 | 42.7 | 30.9 | 69.4 | 37.6 | 51.0 | 45.54 |
| sGLA | 17.63 | 18.06 | 30.1 | 56.9 | 46.4 | 44.6 | 30.9 | 70.4 | 36.7 | 50.3 | 45.79 |
| GDN | 17.27 | 15.76 | 30.8 | 54.0 | 46.8 | 44.0 | 31.3 | 71.3 | 38.1 | **51.9** | 46.03 |
| RLA (ours) | 17.35 | 15.59 | 30.6 | 56.6 | **48.1** | 46.3 | 31.0 | 70.7 | **38.5** | 49.7 | 46.44 |
| RDN (ours) | **16.57** | **14.93** | **32.1** | **58.7** | 47.7 | **48.7** | **31.6** | **71.7** | 37.7 | 49.5 | **47.20** |

Table 3: Accuracy on recall-intensive benchmarks. The **bold**/underlined numbers indicate the first/second best values of the linear attention models in each column.

| Model | DROP | FDA | NQ | SQD | SWDE | TQA | NIAH | Avg |
|---|---|---|---|---|---|---|---|---|
| Transformer | 26.9 | 63.0 | 29.8 | 35.7 | 68.7 | 43.6 | 70.5 | 48.31 |
| RetNet | 26.2 | 35.3 | 20.8 | 31.5 | 44.1 | 39.9 | 65.7 | 37.64 |
| Mamba2 | 26.2 | 41.2 | 23.6 | 33.0 | 63.4 | 43.2 | 67.2 | 42.54 |
| sGLA | 25.7 | 51.8 | 24.8 | 31.6 | 63.4 | 41.5 | 76.6 | 45.06 |
| GDN | 25.9 | 47.4 | **26.8** | 32.4 | 63.3 | **43.5** | 75.7 | 44.99 |
| RLA (ours) | 26.5 | 51.5 | 25.2 | **33.6** | **64.4** | 42.3 | **83.6** | 46.73 |
| RDN (ours) | **27.8** | **57.5** | 26.3 | 32.5 | 63.4 | 43.1 | 79.2 | **47.11** |

## 4.3 ABLATION STUDY

In this section, we present a series of ablation studies to verify the contributions of key components. We first quantify the advantage of our learned residual fitting approach over a predefined correction. Next, we investigate the importance of using a dedicated correction factor, followed by an analysis of the necessity of the gated mechanism for combining the base prediction and the correction. Finally, we examine the effect of normalization and residual clipping.

**Residual Fitting** To validate the importance of accumulating past errors, we test a variant that uses a simpler, predefined correction term. In this ablation, we replace our persistent auxiliary state, $R_t$, with a stateless correction derived only from the current residual, $R_t = (v_t - S_{t-1}k_t)k_t^\top$. As demonstrated in Table 4, the variant lacking explicit residual fitting underperforms our full method. Although this ablated variant maintains competitive performance on some benchmarks, it exhibits a substantial increase in perplexity on both the training and evaluation sets. This performance drop extends to specialized domains, with a substantial degradation in its math and code abilities, as measured by perplexity on GSM8k (Cobbe et al., 2021) and HumanEval (Chen et al., 2021). This demonstrates the critical role of the auxiliary state in accumulating past residuals to refine the model's output effectively.

Table 4: Ablation study of the residual fitting process, comparing training loss and perplexity across various datasets. All models were pretrained for 50B tokens with the same hyperparameters, and the best results are shown in **bold**.

| | Training loss | WikiText ppl | LAMB ppl | GSM8k ppl | HumanEval ppl |
|---|---|---|---|---|---|
| RLA | **2.22** | **18.76** | 23.44 | **3.92** | **9.61** |
| RLA w/o fitting | 2.26 | 20.19 | **22.50** | 6.85 | 16.23 |

**Dedicated Correction Factor** We analyze the benefit of using a dedicated correction factor, $\gamma$, by comparing our full models against variants where $\gamma$ is tied to the update factor $\beta$. In Figure 3a,

the models with an independent $\gamma$ consistently achieve lower evaluation loss, with the RDN variant showing greater improvement. This trend extends to downstream performance, as demonstrated by the results in Figure 3b, which also show that the dedicated correction factor yields performance gains across multiple benchmarks. Notably, our foundational architecture, which does not require an additional $\gamma$, still marks a notable improvement over the baseline linear attention method.

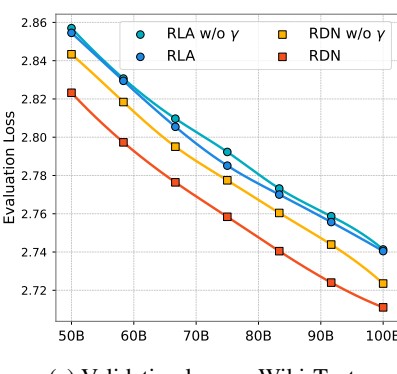

(a) Validation loss on Wiki-Text

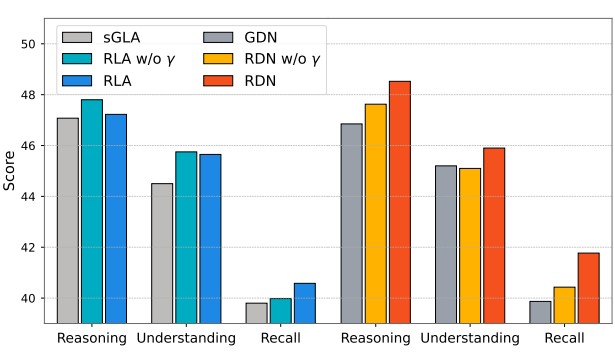

(b) Evaluation result on different types of benchmarks.

Figure 3: Ablation study on the correction factor $\gamma$. Using a dedicated $\gamma$ consistently lowers validation loss compared to tying it to $\beta$. The evaluation uses the same benchmarks as in Section 4.2, divided into three task types, and confirms that a dedicated $\gamma$ improves performance across several categories.

**Gated Output Combination**   We conducted an ablation study to analyze the effect of the output combination formula. This involved comparing our full model, which uses a gated combination of the base prediction and the error correction ($\boldsymbol{o}_t = \alpha_t \boldsymbol{S}_{t-1} \boldsymbol{q}_t + \gamma_t \boldsymbol{R}_t \boldsymbol{q}_t$), against a variant using simple addition ($\boldsymbol{o}_t = \boldsymbol{S}_{t-1} \boldsymbol{q}_t + \boldsymbol{R}_t \boldsymbol{q}_t$). As shown in Table 5, removing the gate causes a slight performance drop for RDN but a slight increase for RLA. This outcome indicates that the core benefit is derived from the residual fitting process on the auxiliary state $\boldsymbol{R}_t$, rather than the specific weight used to integrate the correction term.

Table 5: Ablation study of different output combination methods. All models share the same 50B tokens pretraining and hyperparameters, with the best results for each method shown in **bold**.

| | **Output Combination** | **ARC-C** acc_n | **ARC-E** acc | **HellaSwag** acc_n | **LAMB** acc | **MMLU** acc_n | **PIQA** acc | **SIQA** acc | **Wino** acc | **Avg** |
|---|---|---|---|---|---|---|---|---|---|---|
| RLA | $\alpha_t \boldsymbol{S}_{t-1} \boldsymbol{q}_t + \gamma_t \boldsymbol{R}_t \boldsymbol{q}_t$ | 28.5 | 53.6 | **42.0** | 36.8 | 29.8 | 68.6 | **38.1** | 49.2 | 43.30 |
| | $\boldsymbol{S}_{t-1} \boldsymbol{q}_t + \boldsymbol{R}_t \boldsymbol{q}_t$ | **28.9** | **55.2** | 41.5 | **37.0** | **30.1** | **68.9** | 37.3 | **51.6** | **43.81** |
| RDN | $\alpha_t \boldsymbol{S}_{t-1} \boldsymbol{q}_t + \gamma_t \boldsymbol{R}_t \boldsymbol{q}_t$ | **29.8** | 55.2 | **42.5** | **39.9** | **30.4** | 69.1 | **40.4** | **49.9** | **44.65** |
| | $\boldsymbol{S}_{t-1} \boldsymbol{q}_t + \boldsymbol{R}_t \boldsymbol{q}_t$ | 27.8 | **56.3** | 41.9 | 39.7 | 29.4 | **70.0** | 37.7 | 48.9 | 43.96 |

**Normalization and Residual Clipping**   Finally, we investigate the importance of normalization and residual clipping. We perform an ablation study on RLA by removing normalization and clipping. As shown in Figure 4, both components are crucial for stable training; their removal leads to unbounded activations and degraded performance. In contrast, the RDN model is largely insensitive to residual clipping. This robustness is attributable to the inherent stability of its delta rule update, which maintains a consistent loss curve without residual clipping (Figure 4b).

## 5   RELATED WORKS

Sequence modeling has been historically dominated by Recurrent Neural Networks (RNNs) (Lipton et al., 2015), including variants like Long Short-Term Memory (LSTM) (Hochreiter & Schmidhuber, 1997) and Gated Recurrent Units (GRU) (Cho et al., 2014). While effective, their inherently sequential nature impedes training parallelization. The Transformer architecture (Vaswani et al.,

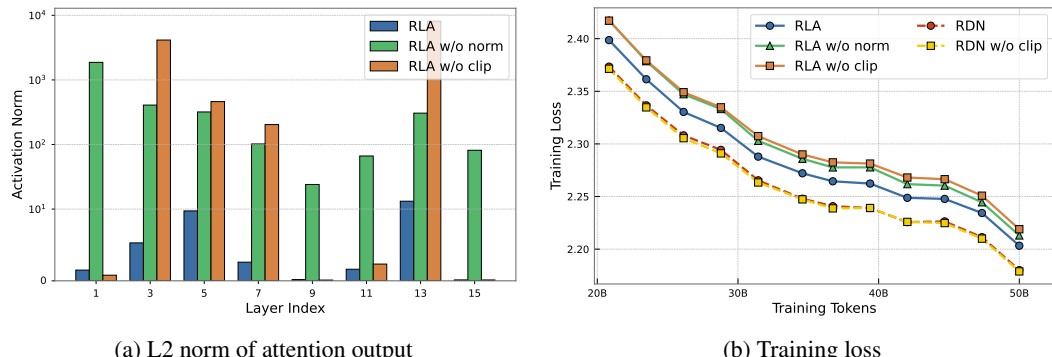

(a) L2 norm of attention output  (b) Training loss

Figure 4: Ablation study on normalization and residual clipping. RLA variants without normalization or clipping exhibit exploding activation norms, indicating training instability. This instability leads to a higher training loss, highlighting that both components are crucial for stable training and better performance. In contrast, residual clipping has a negligible impact on the RDN training process.

2017) overcame this limitation, emerging as the de facto standard for sequence modeling. However, its self-attention mechanism, with a computational complexity quadratic in sequence length, poses a significant bottleneck for long-context applications.

To address these challenges, recent research has revisited Linear RNNs as a foundation for efficient Transformer alternatives. By formulating sequence processing as a linear recurrence, these models achieve both parallelizable training and linear-time inference. Early explorations in this domain, such as S4 (Gu et al., 2021), LRU (Orvieto et al., 2023), and RetNet (Sun et al., 2023), utilized structured state transition matrices. A subsequent performance leap was achieved by incorporating data-dependent dynamics. Models like Mamba (Gu & Dao, 2023; Dao & Gu, 2024), HGRN (Qin et al., 2023; 2024b), and Gated Linear Attention (Yang et al., 2023) leverage input-dependent gating to dynamically control state transitions, thereby enhancing their expressive power.

More advanced methods have introduced the delta learning rule, which reframes the state update from a simple gated decay to a fine-grained memory correction. This approach, exemplified by DeltaNet (Yang et al., 2024b; Schlag et al., 2021) and Gated DeltaNet (Yang et al., 2024a), enables more precise dynamic memory modifications. This mechanism can be interpreted from an online learning perspective, where the state update is framed as an optimization process, as explored in TTT (Sun et al., 2024). This viewpoint has inspired further work aimed at discovering and improving the intrinsic learning algorithms within sequence models (von Oswald et al., 2023; 2025).

Concurrent research has focused on increasing the expressivity of the state transition. For instance, RWKV-7 (Peng et al., 2025) employs a diagonal-plus-low-rank structure, while DeltaProduct (Siems et al., 2025) generalizes DeltaNet by performing multiple update steps per token. To push capacity even further, recent architectures such as Titans (Behrouz et al., 2024) and Miras (Behrouz et al., 2025) have introduced non-linear deep memory, parameterizing the state with an MLP.

## 6 CONCLUSION

In this paper, we introduced Residual Linear Attention, a framework that enhances linear attention models with an explicit residual fitting process. Our method leverages an auxiliary state to correct the predictive errors of the base model, thereby building more robust and accurate contextual representations. The framework is highly adaptable and can be applied to various linear attention methods. Our experiments demonstrated this versatility, showing that our approach consistently outperforms their respective baselines. While this improvement comes at the cost of additional computation for the fitting process, balancing this trade-off offers a promising direction for future research.

## REPRODUCIBILITY STATEMENT

To ensure the reproducibility of our research, we will make our resources publicly available. The complete source code is provided in the supplementary material. Our evaluation is conducted using the `lm-evaluation-harness` (Gao et al., 2024) framework to ensure fair and consistent comparison with prior work. Details regarding our experimental setup are described in Section 4.1, and a list of model parameters and hyperparameters can be found in Appendix C.

## USE OF LARGE LANGUAGE MODELS

We would like to thank Google's Gemini for its assistance in editing this manuscript. Its suggestions were used to improve grammar, spelling, and overall clarity. The authors reviewed all modifications and take full responsibility for the content of this paper.

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

# A UNIFIED FORMULA FOR RESIDUAL LINEAR ATTENTION

This section details how our residual fitting process can be framed as a form of online gradient boosting, providing a unified and extensible framework.

## A.1 PRELIMINARY: GRADIENT BOOSTING IN A FUNCTIONAL SPACE

Gradient boosting is an ensemble technique that sequentially adds new models to correct the errors of previous ones. In a gradient boosting framework, the objective at each stage is to find a new function $h$ that minimizes the loss when added to the current function $f$:

$$h^* = \arg\min_h \mathbb{E}_{\boldsymbol{k}, \boldsymbol{v}}[\mathcal{L}(f(\boldsymbol{k}) + h(\boldsymbol{k}), \boldsymbol{v})].$$

As finding the optimal function $h$ is generally infeasible, we approximate the objective using a first-order Taylor expansion of the loss:

$$\mathcal{L}(f(\boldsymbol{k}) + h(\boldsymbol{k}), \boldsymbol{v}) \approx \mathcal{L}(f(\boldsymbol{k}), \boldsymbol{v}) + h(\boldsymbol{k})\frac{\partial \mathcal{L}(f(\boldsymbol{k}), \boldsymbol{v})}{\partial f(\boldsymbol{k})}.$$

The direction of steepest descent in this functional space is the negative gradient of the loss with respect to the function's output. This target is often called a pseudo-residual, $\boldsymbol{r}$:

$$\boldsymbol{r} = -\frac{\partial \mathcal{L}(f(\boldsymbol{k}) + h(\boldsymbol{k}), \boldsymbol{v})}{\partial h(\boldsymbol{k})} \approx -\frac{\partial \mathcal{L}(f(\boldsymbol{k}), \boldsymbol{v})}{\partial f(\boldsymbol{k})}.$$

The objective thus becomes learning a function $h(\boldsymbol{k})$ that fits this pseudo-residual. After learning, the boosted function is updated as $f(\boldsymbol{k}) \leftarrow f(\boldsymbol{k}) + h(\boldsymbol{k})$, where $h(\boldsymbol{k})$ is the error correction term.

## A.2 UNIFIED RESIDUAL LINEAR ATTENTION

From an online learning perspective, linear attention can be viewed as learning a mapping $f(\boldsymbol{k}; \boldsymbol{S})$, where the state matrix $\boldsymbol{S}$ is incrementally updated via online gradient descent to minimize a loss $\mathcal{L}(f(\boldsymbol{k}; \boldsymbol{S}), \boldsymbol{v})$. We enhance this model by incorporating principles from gradient boosting. This involves employing an auxiliary state matrix $\boldsymbol{R}$ for iterative refinement. In this framework, state matrix $\boldsymbol{R}$ is updated at timestep $t$ to learn the mapping from the key $\boldsymbol{k}_t$ to the pseudo-residual $\boldsymbol{r}_t$ of the prior prediction. This correction process results in a stronger mapping function.

We can decouple the learning objective into two parts: (1) A global objective, defined by an arbitrary, differentiable outer loss $\mathcal{L}_{\text{outer}}$, which sets the overall key-to-value mapping goal. (2) A local objective, defined by a simple inner loss $\mathcal{L}_{\text{inner}}$, which governs how each individual state matrix is updated. While the target pseudo-residual $\boldsymbol{r}$ can be complex, the task for each state is deliberately kept simple. Ignoring decay factors and learning rates for clarity, the general recurrence is:

$$\boldsymbol{r}_t = -\frac{\partial \mathcal{L}_{\text{outer}}(f(\boldsymbol{k}_t; \boldsymbol{S}_{t-1}), \boldsymbol{v}_t)}{\partial f(\boldsymbol{k}_t; \boldsymbol{S}_{t-1})} \qquad \text{(Pseudo-Residual)}$$

$$\boldsymbol{R}_t = \boldsymbol{R}_{t-1} - \frac{\partial \mathcal{L}_{\text{inner}}(f(\boldsymbol{k}_t; \boldsymbol{R}_{t-1}), \boldsymbol{r}_t)}{\partial \boldsymbol{R}_{t-1}} \qquad \text{(Auxiliary State Update)}$$

$$\boldsymbol{S}_t = \boldsymbol{S}_{t-1} - \frac{\partial \mathcal{L}_{\text{inner}}(f(\boldsymbol{k}_t; \boldsymbol{S}_{t-1}), \boldsymbol{v}_t)}{\partial \boldsymbol{S}_{t-1}} \qquad \text{(Base State Update)}$$

$$\boldsymbol{o}_t = f(\boldsymbol{q}_t; \boldsymbol{S}_{t-1}) + f(\boldsymbol{q}_t; \boldsymbol{R}_t) \qquad \text{(Base Prediction and Correction)}$$

The gating mechanism and correction strength can also be easily incorporated into the framework. This framework allows two simple inner update rules to approximate a more complex global objective.

# B RESIDUAL FITTING WITH HUBER LOSS

As previously established, our framework can accommodate complex global loss functions, as their complexity is confined to the pseudo-residual calculation. This allows us to use a more robust alternative to the standard L2 loss, such as the Huber loss function:

$$\mathcal{L}_{\text{huber}}(\hat{\boldsymbol{v}}, \boldsymbol{v}) = \sum_{i=1}^{d} \mathcal{L}_{\text{huber}}(\hat{v}_i, v_i),$$

$$\text{where } \mathcal{L}_{\text{huber}}(\hat{v}_i, v_i) = \begin{cases} \frac{1}{2}(v_i - \hat{v}_i)^2 & \text{for } |v_i - \hat{v}_i| \leq c \\ c\left(|v_i - \hat{v}_i| - \frac{1}{2}c\right) & \text{for } |v_i - \hat{v}_i| > c \end{cases}$$

This function uses the L2 loss for small errors and the L1 loss for large errors, making it a more robust alternative. Directly applying this loss yields a non-linear update rule that is difficult to parallelize:

$$\boldsymbol{S}_t = \boldsymbol{S}_{t-1} - \frac{\partial \mathcal{L}_{\text{huber}}(\boldsymbol{v}_t - \boldsymbol{S}_{t-1}\boldsymbol{k}_t)}{\partial \boldsymbol{S}_{t-1}}$$

$$= \boldsymbol{S}_{t-1} + \text{Clip}_{[-c,c]}(\boldsymbol{v}_t - \boldsymbol{S}_{t-1}\boldsymbol{k}_t)\boldsymbol{k}_t^{\top}$$

Our residual fitting framework elegantly avoids this problem. The complexity is isolated within the pseudo-residual calculation, while the core state update remains simple. The pseudo-residual for the Huber Loss is $\boldsymbol{r}_t = \text{Clip}_{[-c,c]}(\boldsymbol{v}_t - f(\boldsymbol{k}_t; \boldsymbol{S}_{t-1}))$, then the inner update rule is only responsible for fitting this target pseudo-residual. Using an inner loss of $\mathcal{L}_{\text{inner}}(f(\boldsymbol{k}), \boldsymbol{v}) = -\langle \boldsymbol{v}, f(\boldsymbol{k}) \rangle$ yields the following recurrence:

$$\boldsymbol{r}_t = \text{Clip}_{[-1,1]}(\boldsymbol{v}_t - \boldsymbol{S}_{t-1}\boldsymbol{k}_t)$$

$$\boldsymbol{R}_t = \boldsymbol{R}_{t-1} - \boldsymbol{r}_t \boldsymbol{k}_t^{\top}$$

$$\boldsymbol{S}_t = \boldsymbol{S}_{t-1} - \boldsymbol{v}_t \boldsymbol{k}_t^{\top}$$

This equivalence provides the theoretical motivation for our clipping mechanism; it is an efficient implementation of a robust Huber loss objective, which leads to a more stable residual fitting process. This principle can be generalized by selecting other robust loss functions. For instance, the Log-Cosh loss has a negative gradient with respect to the prediction $f(\boldsymbol{k})$ that is equivalent to applying a $\tanh$ function to the residual:

$$\mathcal{L}_{\text{log-cosh}}(f(\boldsymbol{k}), \boldsymbol{v}) = \log \cosh(\boldsymbol{v} - f(\boldsymbol{k})),$$

$$\boldsymbol{r}_t = \tanh(\boldsymbol{v}_t - f(\boldsymbol{k}_t; \boldsymbol{S}_{t-1})).$$

This can be viewed as a smooth alternative to the clipped residual from the Huber loss. The ability to easily substitute such loss functions demonstrates our framework's modularity, allowing for the integration of powerful learning objectives while maintaining computational efficiency.

# C MODEL STRUCTURE AND TRAINING HYPER PARAMETERS

We evaluate several model architectures, with full specifications detailed in Table 6. Our comparison includes a standard Transformer and Transformers with pure linear attention. For a fair comparison, all models are trained using an identical set of hyperparameters, which are listed in Table 7. We initialize the model weights from a normal distribution with a constant standard deviation and use the AdamW optimizer with a cosine learning rate schedule for training.

Table 6: Model Configuration

| Property | Transformer Models | Linear Attention Models |
|---|---|---|
| Total Params | 1.51B | 1.55B |
| Hidden Size | 2048 | 2048 |
| Intermediate Size | 8192 | 8192 |
| Attention Heads | 32 | 16 |
| GQA Groups | 4 | 16 |
| Head Dimension | 128 | 128 |
| Softmax Attention Layers | 16 | 0 |
| Linear Attention Layers | 0 | 16 |

Table 7: Training hyperparameters

| Peak LR | Min LR | Batch Size | Warmup Tokens | Total Tokens | Weight Decay | Gradient Clip | Initialization Std |
|---|---|---|---|---|---|---|---|
| 3e-4 | 3e-5 | 4M | 0.5B | 100B | 0.1 | 1.0 | 0.006 |

## D PSEUDO-CODE IMPLEMENTATION

This section provides a PyTorch-like pseudo-code implementation for our proposed Residual Linear Attention (RLA). We present the recurrent formulation to clearly illustrate our modifications to a baseline scalar-gated linear attention mechanism.

```python
# q, k, v are in shape [sequence_length, head_dimension].
# alpha, beta and gamma are in shape [sequence_length]

def scalar_gated_linear_attention(q, k, v, alpha, beta):
    # Recurrently compute linear attention with scalar gates.
    seq_len, head_dim = q.shape
    S = torch.zeros(head_dim, head_dim)
    o = torch.zeros(seq_len, head_dim)
    for t in range(seq_len):
        qt, kt, vt = q[t : t + 1], k[t : t + 1], v[t : t + 1]
        # update state S
        S = alpha[t] * S + beta[t] * kt.T @ vt
        # get prediction
        o[t : t + 1] = qt @ S
    return o

def residual_linear_attention(q, k, v, alpha, beta, gamma):
    # Recurrently compute residual linear attention.
    seq_len, head_dim = q.shape
    S = torch.zeros(head_dim, head_dim)
    R = torch.zeros(head_dim, head_dim)
    o = torch.zeros(seq_len, head_dim)
    for t in range(seq_len):
        qt, kt, vt = q[t : t + 1], k[t : t + 1], v[t : t + 1]
        # l2 normalization
        qt, kt = F.normalize(qt, dim=-1), F.normalize(kt, dim=-1)
        # clipped residual error
        rt = torch.clip(vt - kt @ S, min=-1, max=1)
        # update auxiliary state R
        R = alpha[t] * R + gamma[t] * kt.T @ rt
        # combine basic prediction and error correction
        o[t : t + 1] = alpha[t] * qt @ S + gamma[t] * qt @ R
        # update basic state S
        S = alpha[t] * S + beta[t] * kt.T @ vt
    return o
```

Listing 1: Pseudo-code for Residual Linear Attention (RLA) and a baseline linear attention model.

