# OpenReview forum: "Enhancing linear attention with residual learning"
_ICLR.cc/2026/Conference — ICLR 2026 Conference Withdrawn Submission_

### Official Review · Reviewer_7tVn · 2025-10-21

**Soundness:** 2
**Presentation:** 3
**Contribution:** 2
**Rating:** 4
**Confidence:** 4

**Summary:**

This paper introduces a novel view of the state update mechanism in linear attention, framing it as a "prediction-correction" process. By analyzing current methods, the authors argue that previous approaches adopt a correction term based only on the current input token. To address this limitation, Residual Linear Attention (RLA) introduces an auxiliary RNN state ($R_t$). This auxiliary state accumulates historical errors, making the correction term relevant to past tokens, not just the current one. Experiments show that this RLA framework enhances the performance of baseline linear models.

**Strengths:**

The paper is clearly written and identifies an interesting problem in existing linear attention models: the reliance on a single-token correction term . The proposed RLA framework is a structured attempt to address this by introducing an auxiliary state to accumulate historical errors . The method demonstrates empirical improvements over its direct baselines (sGLA and GDN) across the presented language modeling and recall tasks.

**Weaknesses:**

1. The theoretical analysis of residual learning in linear attention is not particularly sound. In my opinion, the newly introduced RNN state, $R$, can be mathematically reformulated as a separate state instead of being truly coupled with the primary hidden state $S$. **The residual learning procedure thus reduces to a summation of the outputs of two separate linear attention computations.** I explain this below:

The authors state:
\begin{align}
r_t &= v_t - S_{t-1}k_t, \\\\
R_t &= \alpha_tR_{t-1} + \gamma_tr_tk_t^T, \\\\
S_t &= \alpha_tS_{t-1} + \beta_tv_tk_T^T, \\\\
o_t &= (\alpha_tS_{t-1} + \gamma_tR_t)q_t
\end{align}
If we treat the combined state for the output as $M_{t-1} = \alpha_{t-1}S_{t-2} + \gamma_{t-1}R_{t-1},$
then for RLA, we have
\begin{align}
M_t &= \alpha_tS_{t-1} + \gamma_t(\alpha_tR_{t-1} + \gamma_tr_tk_t^T) \\\\
&= \alpha_tS_{t-1} + \gamma_t(\alpha_tR_{t-1} + \gamma_t(v_t-S_{t-1}k_t)k_t^T) \\\\
&= S_{t-1}(\alpha_tI - \gamma_t^2k_tk_t^T) + \alpha_t\gamma_tR_{t-1} + \gamma_t^2v_tk_t^T.
\end{align}
EThis derivation shows that the update for $M$ can be separated into two distinct recurrent updates, which implies that the explicit correlation between $S$ and $R$ is not necessary. The update of $R$ based on the so-called residual coupling with $S$ can be absorb into the update of $S$ itself to form a new, independent recurrent state..

In its original form, RLA is presented as a GLA hidden state with a coupled GLA correction state. However, it is mathematically equivalent to a summation of a GDN hidden state and a GLA hidden state. The output is equivalent to computing two separate GDN and GLA states and then summing them.

For RDN, we have
\begin{align}
M_t &= \alpha_tS_{t-1} + \gamma_t(\alpha_tR_{t-1}(I-\gamma_tk_tk_t^T) + \gamma_tr_tk_t^T) \\\\
&= \alpha_tS_{t-1} + \gamma_t(\alpha_tR_{t-1}(I-\gamma_tk_tk_t^T) + \gamma_t(v_t-S_{t-1}k_t)k_t^T) \\\\
&= \alpha_tS_{t-1} + \alpha_t\gamma_tR_{t-1}(I-\gamma_t k_tk_t^T) + \gamma_t^2 v_tk_t^T - \gamma^2_tS_{t-1}k_tk_t^T \\\\
&= S_{t-1}(\alpha_tI-\gamma^2_tk_tk_t^T) + \alpha_t\gamma_tR_{t-1}(I-\gamma_t k_tk_t^T) + \gamma_t^2 v_tk_t^T.
\end{align}
This means that RDN, stated as a GDN hidden state with a coupled GDN correction state, is mathematically equivalent to a summation of two independent GDN hidden states.

As for the computational cost, RLA doubles the hidden state size and thus roughly doubles the computational cost. This aligns with the argument that the method is equivalent to computing two separate LAs and summing their results.

2. Even if the RLA can be explained as a strongly coupled learning method, the doubled hidden states is not fair for other baselines presented in the paper. An important baseline is that compute two separate types of LAs (e.g., one GDN and one GLA, or two GDNs) and sum them up, which is missing.

**Questions:**

1. RLA needs to call the FLA operator twice to update S and R respectively. As far as I know, the time overhead for launching an operator is significant, and obtaining the intermediate state to perform norm and clip operations will also increase computational overhead. How can you explain why RLA's speed is only slightly lower than the original operator?
2. The idea of using a second-order Taylor expansion to explain the residual is good, but I don't understand where the second-order gradient is reflected.

---

### Official Review · Reviewer_YePu · 2025-10-30

**Soundness:** 3
**Presentation:** 3
**Contribution:** 3
**Rating:** 6
**Confidence:** 4

**Summary:**

This paper introduces a new framework, Residual Linear Attention, to address a key limitation in existing linear attention models: an expressivity bottleneck where the model's correction mechanism relies solely on the current token. The authors propose enhancing linear attention with an explicit residual-fitting mechanism. Experiments on language modeling (PPL) and recall-intensive benchmarks (e.g., NIAH) show that RLA and RDN consistently outperform their respective baselines (sGLA and GDN) and other modern linear-time models, thereby narrowing the performance gap with standard Transformers. The paper also provides theoretical justification for its residual clipping mechanism as an efficient implementation of a robust Huber loss objective

**Strengths:**

- The primary strength is the novel and clearly-articulated residual-fitting framework. The idea of using an auxiliary state $R_t$ to explicitly accumulate past residuals $r_t$  is an intuitive and original extension of the online learning view of linear attention. The decomposition of the output into a "base prediction" and a "learned correction" (Section 2.3)  provides exceptional clarity and motivation.

- The empirical validation is good. The authors perform the correct apples-to-apples comparisons: RLA against its sGLA baseline and RDN against its GDN baseline. The consistent gains shown across a wide variety of tasks, from standard PPL benchmarks to difficult recall-intensive (NIAH) and reasoning tasks, demonstrate the robustness and effectiveness of the proposed mechanism.

- The paper includes a comprehensive set of ablation studies in Section 4.3. These studies successfully validate the importance of the core residual fitting process, the dedicated correction factor $\gamma$ , and the stability mechanisms (clipping/normalization), adding significant weight to the paper's design choices.

**Weaknesses:**

- A key weakness is the apparent instability of the base RLA variant. The ablation study in Section 4.3 (Figure 4) shows that RLA's training is unstable without L2 normalization or residual clipping, suffering from "unbounded activations". While RDN is shown to be robust, the fact that one of the two main proposed models has this fragility is a concern. It suggests that the simple additive residual update may be inherently unstable and relies on these additional components to function, making it less robust than the RDN variant.

- The paper's own ablation study calls into question the necessity of the gated output combination ($o_{t}=\alpha_{t}S_{t-1}q_{t}+\gamma_{t}R_{t}q_{t}$). In Table 5, the RLA variant using simple addition ($o_t = S_{t-1}q_t + R_t q_t$) actually achieves a higher average accuracy (43.81) than the full RLA model with the more complex gated output (43.30). The authors even conclude that the core benefit stems from the $R_t$ state itself, not the specific combination method. This makes the inclusion of the $\alpha_t$ and $\gamma_t$ output gates seem like unnecessary complexity, at least for the RLA model.

**Questions:**

Please see my weakness part.

---

### Official Review · Reviewer_5d45 · 2025-11-03

**Soundness:** 1
**Presentation:** 2
**Contribution:** 2
**Rating:** 2
**Confidence:** 4

**Summary:**

This paper proposes Residual Linear Attention (RLA) as an enhanced linear attention variant. The linear attention can be decomposed into a base prediction and an error correction. RLA maintains an auxiliary recurrent state to accumulate residual errors and correct the base prediction. For the efficiency, RLA introduces minor computation cost. The experiments show that RLA achieves performance improvement over a selection of benchmarks.

**Strengths:**

1. This paper proposes a novel linear attention variant, which stabilizes LLMs' pre-training process.

2. The proposed residual learning method introduce minor computation overhead.

3. Experiments span a series of different tasks, LM, reasoning, and some recall-intensive tasks.

**Weaknesses:**

1. **limited performance gain and increased computation** The proposed RLA introduces additional computational overhead compared to GDA (Gated DeltaNet), yet it does not demonstrate a pronounced improvement in performance.

2. **Unclear pre-training setup and model size selection** The experiments are based on 1.5B models pre-trained using 100B tokens. First, it is unclear the models are pre-trained from scratch or CPT from existing checkpoints. If pre-trained from scratch, the selected model is relatively small in size. LLMs are typically considered to start from around 7B parameters. More importantly, 100B tokens are not sufficient to fully pre-train a 1.5B LLM, which means the model has not converged during the training process. The consequent performance comparison can be unfair. If the model is CPT, 1.5B model remains rather small and may not serve as a strong representative of LLM behavior.

3. **Insufficient experimental details** The experimental settings are significantly insufficient for readers to assess the soundness of the experimental results. For example, the paper does not specify the pre-training corpus used, nor does it report training and validation losses across different model architectures. They are very important indicators to assess and supervise the pre-training process. In addition, it is unclear how many runs were performed for each experiment. One can easily sweep on random seeds to achieve similar numbers.

4. **The design of RLA are not well ablated and verify the effectiveness.** For example, from Table 5, the adaptive gating does not appear to contribute noticeable improvement.

**Questions:**

Please see above.

---

### Official Review · Reviewer_9q9Q · 2025-11-03

**Soundness:** 2
**Presentation:** 2
**Contribution:** 2
**Rating:** 4
**Confidence:** 3

**Summary:**

The paper proposes Residual Linear Attention (RLA), a framework that enhances linear attention mechanisms by maintaining an auxiliary recurrent state to explicitly model and correct prediction errors.

The key idea is to learn residuals $r_t = v_t - S_{t-1}k_t $ and accumulate them in a secondary state matrix $R_t$ to improve predictions. Two variants are presented: RLA (with additive updates) and RDN (with delta-rule updates).


Experiments on 1.5B parameter models show improvements over baselines on language modeling and recall tasks.

The core contribution is adding a second recurrent state that learns residuals—essentially two parallel linear attention mechanisms, one for prediction errors. Prior work DeltaNet (Yang et al., 2024b) already computes $R_t = (v_t - S_{t-1}k_t)k_t^\top$ each step; RLA/RDN merely make it stateful via $R_t = \alpha_tR_{t-1} + \gamma_tr_tk_t^\top$, an engineering change rather than a conceptual one.

**Strengths:**

- The idea of maintaining a separate recurrent state for residual corrections is interesting and shows empirical gains.
-  Evaluation spans multiple benchmarks (WikiText, reasoning tasks, recall-intensive tasks) with consistent improvements.
- Section 4.3 systematically validates design choices (correction factor $\gamma$, clipping, normalization).

**Weaknesses:**

1. The decomposition $o_t = S_{t-1}q_t + R_tq_t$ conflates fundamentally different concepts. For standard linear attention, $R_t = v_tk_t^\top$ is not a persistent state matrix but represents $v_t(k_t^\top q_t)$—a scalar-weighted vector recomputed each step. In contrast, the proposed method maintains $R_t$ as a recurrent state accumulating information over time. Table 1 obscures this difference between stateless corrections and stateful accumulation. The “unified framework” claim (lines 124–130) is overstated, listing different methods without real unification or theoretical insight.

2. The second-order Taylor expansion argument (lines 153–159) is unnecessarily complex. For L2 loss $L = \tfrac{1}{2}\|v - \hat{v}\|^2$, the optimal update $\delta^\star = v - \hat{v}$ follows directly from $\nabla_{\hat{v}}L = \hat{v} - v = 0$. The second-order term $\nabla^2_{\hat{v}} L = I$ adds no insight since the Hessian is identity.

3. I think Appendix B may a theoretical error: it claims residual clipping “is an efficient implementation of a robust Huber loss” (line 753). Is this correct? True Huber loss would update
$
S_t = S_{t-1} +  \text{Clip}(v_t - S_{t-1}k_t)k_t^\top,
$
but the implementation clips only the auxiliary residual $r_t = \text{Clip}(v_t - S_{t-1}k_t)$ affecting $R_t$, while $S_t = \alpha_t S_{t-1} + \beta_t v_t k_t^\top$ remains unclipped. Thus, clipping stabilizes residual learning but is not Huber-equivalent.



4. The method doubles recurrent state memory, maintaining $S_t, R_t \in \mathbb{R}^{d_v \times d_k}$. With 16 heads, $d_v = d_k = 128$, and 16 layers, each state matrix adds roughly 32 MB, totaling about 1 GB extra memory—substantial for long-context inference. Figure 2 reports runtime but omits GPU memory, computational overhead (lines 267–268), and memory–performance tradeoffs, limiting assessment of efficiency.

5. The paper cites TTT-MLP (Sun et al., 2024), Titans (Behrouz et al., 2024), and Miras (Behrouz et al., 2025) as sacrificing linear recurrence, but provides no comparison. Given that the proposed method also increases memory (2× states) and computation, and Miras claims to retain recurrence through test-time memorization, the distinction is unclear. Section 5 lists these works without analyzing tradeoffs between state complexity (single vs dual vs MLP) and performance.

6. The core contribution is adding a second recurrent state that learns residuals--essentially two parallel linear attention mechanisms, one for prediction errors. Prior work DeltaNet (Yang et al., 2024b) already computes $R_t = (v_t - S_{t-1}k_t)k_t^\top$ each step; RLA/RDN merely make it stateful via $R_t = \alpha_tR_{t-1} + \gamma_tr_tk_t^\top$, an engineering change rather than a conceptual one.

7. Improvements over Gated DeltaNet (GDN) are modest: 1-2 percent improvement on recall tasks, with doubled memory cost. Comparisons with the custom sGLA baseline are also marginal, and stronger gains appear mainly against older baselines such as Mamba2 and RetNet.

8. The lack of error bars, seed variance, or confidence intervals weakens statistical validity. Ablations use 50 B tokens, and the clipping threshold $c = 1$ is fixed without exploration. Missing ablations include alternative $c$ values, low-rank $R_t$, mixed update rules, model-size scaling, and initialization of $R_0$.

9. The link to gradient boosting might be weak. Classical boosting trains models sequentially, but here $S_t$ and $R_t$ update in parallel. The “unified formula” in Appendix A.2 (lines 688–696) restates existing updates without new methods or insights. Despite claiming generality, the framework yields only two straightforward variants (RLA, RDN) and no new design principles beyond adding a residual state.

**Questions:**

Please see Weaknesses.

---

### Note · Authors · 2025-12-02

I have read and agree with the venue's withdrawal policy on behalf of myself and my co-authors.